# Psychological, Psychosocial and Obstetric Differences between Spanish and Immigrant Mothers: Retrospective Observational Study

**DOI:** 10.3390/ijerph191811782

**Published:** 2022-09-18

**Authors:** María Carmen Martínez Herreros, María Fe Rodríguez Muñoz, Nuria Izquierdo Méndez, María Eugenia Olivares Crespo

**Affiliations:** 1Department of Psychology, National University of Distance Education (UNED), C/Juan del Rosal No. 10, 28040 Madrid, Spain; 2Department of Obstetrics and Gynecology, San Carlos Clinic Hospital, 28040 Madrid, Spain

**Keywords:** prenatal mental health, immigration, sociodemographic factors, obstetric problems, birth problems

## Abstract

This study analyzed the influence of psychological and psychosocial factors of pregnant women at an obstetric level. The possible differences between Spaniards and immigrants were studied. This was a retrospective observational study. The sample has been divided into two study cohorts, one consisting of Spanish pregnant women and one consisting of foreign pregnant women. Both completed the Revised Postpartum Depression Predictors Inventory and the Patient Health Questionnaire-9. A total of 15.9% of Spanish women and 23.2% of immigrants had depressive symptoms. Immigrants claim to have less support at the partner, family, and friendship levels than Spaniards. Moreover, 16.4% of Spaniards vs. 8.1% of immigrants had pregnancy complications; Cesarean section was performed in 16.2% of Spaniards vs. 7.9% of immigrants. A greater number of premature births were detected in immigrants than in Spaniards. Access to universal healthcare is a protective factor against socioeconomic and cultural conditions affecting the mental and obstetrical health of immigrants.

## 1. Introduction

Antenatal depression can become a real public health problem [1], just like psychosocial problems [2], so their prediction and prevention must be a relevant objective of public health agendas. It is important, due to the repercussions that they can have on the newborn, to check if immigrant pregnant women, who usually present worse psychosocial indicators, have worse obstetric results.

### 1.1. Perinatal Depression

Pregnancy is a process that involves significant physical and emotional changes. Even in an uncomplicated pregnancy, changes that occur can affect both maternal mental health and the development of the baby [3]. Perinatal maternal disorders have profound and pervasive effects on the mother and the mother–infant relationship [4].

Perinatal depression is widely recognized as a serious mental health problem in developed countries and, more recently, in low and middle-income countries [4].

Pregnant women may have difficulty recognizing their depressive symptoms during pregnancy and postpartum because they may be confused with the distress associated with the transition to motherhood. It can also be due to feelings of guilt, shame and stigma, which leads them to hide how they feel [1].

For this reason, the American College of Obstetricians and Gynecologists [5] and the NICE guide [6] indicate the importance of detecting possible risks, not only obstetric and clinical, but also psychosocial (early detection of violence, economics, social and family support, etc.), to improve the health and quality of life of pregnant women and newborns [7,8]. The aim is to identify women who are vulnerable to affective problems during pregnancy and carrying out preventive actions or initiate appropriate treatments [7,8].

### 1.2. Sociodemographic Factor

The first sociodemographic factor to be studied is migration. In recent decades, there has been extensive female migration to developed countries. This phenomenon highlights the differences between women who migrate: family reunification, the migration of single women seeking economic improvement over work at home and care or work in other more highly qualified sectors [9], and refugees; this situation can sometimes become a stressor due to the socio-family burden [10], a poor economic situation, discrimination or social exclusion [11], leading to adaptive and mental health disorders [10,11]. Migrant women are a heterogeneous group, as they depend on the length of stay in the country of destination, the documentation and the condition of residence, whether migration is forced or voluntary and the reasons for migration [11]. Mainly, the migrant population is a young population, of childbearing age, which has a significant impact on national birth rate and the profile of pregnant women [11,12].

### 1.3. Psychosocial Factors and Immigration

The second factor of study is psychosocial health. Psychosocial health is multidimensional and refers to the interrelationships of the social environment, e.g., economic, educational, social support and the psychological health of an individual (such as depression, anxiety, etc.) [2]. A meta-analysis conducted in England [13] shows the determining differences for mental and physical health between immigrants and natives. Many immigrants experience poverty when they arrive in the country of destination. There is also a low level of social support due to the separation of social networks due to migration and social isolation in the host country [13]. Other studies comparing English-speaking women with non-English-speaking women [14,15] indicate that women with language barriers have difficulty accessing maternity services to arrange appointments or seek safe and appropriate care due to lack of knowledge of the operation of health services, as well as cultural barriers, which may lead to their health needs not being met or even to not having access to health services [10,11].

It is important to have access to healthcare. Having continuous access to the free care that some countries provide is not the same as having to pay for healthcare and transportation [11]. This is not only the case in Europe. In the United States, where immigration policies are implemented and delegated to state jurisdictions, these policies may affect the access and use of public health services, which may lead to a delayed and inadequate prenatal care [14]. As a result, immigrant women may be at higher risk of maternal and neonatal morbidity and mortality compared to native women [11].

### 1.4. Obstetric Problems

Other factors of study are the obstetric problems arising from psychosocial problems. One of the consequences of some of these problems is low birth weight, including an increased risk of mortality, health and developmental problems for infants and children [16,17]. This situation bears emotional and economic costs for families; however, the evidence is not yet definitive as to whether antenatal depression is a clear risk factor for PP (prematurity) and LBW (low birth weight) [17].

## 2. Materials and Methods

### 2.1. Design

Most works are performed with samples where there is usually no universal care for pregnant women. The objective of this study in Spain, where there is universal access to healthcare, is to determine which psychosocial and psychological factors are involved in pregnant woman and how they influence the mother and the baby at an obstetric level. We search for obstetrical differences between Spanish women and immigrant.

### 2.2. Settings

This is a retrospective observational study in which data were collected and used to compare the two groups of women: Spaniards and immigrants. For the collection of objective data, the history of pregnancy and childbirth was used, which was provided by the obstetrics department after the study was approved by the Ethics and Clinical Research Committee of the Hospital. The medical history is key to controlling pregnancy [6,18]. In this study, the medical records contain data on both the mother’s pregnancy and the baby’s health at the time of birth. The data collection, due to its confidentiality, was performed in the same hospital.

### 2.3. Participants

A total of 764 women were randomly selected to participate in the investigation of mothers and babies in the Gynecology and Obstetrics Service of the “San Carlos Clinic Hospital”, where their pregnancy was attended to, between the years 2015 and 2017.

Figure 1 gives and overview of the numbers of participants in the research, excluding women who did not meet the requirements.

### 2.4. Procedure

When the women attended an obstetric consultation at 12 weeks of pregnancy for the first trimester ultrasound, they completed a questionnaire composed of the PDPI-R and the PHQ-9 (the aim was to find out which psychosocial risk factors and symptoms of depression could have an influence on the obstetric variables). The pregnant women were informed of the purpose of the research, and those who participated did so voluntarily, giving their written consent.

The obstetric variables (pregnancy and delivery) were collected from the medical records of the mothers who participated in the study.

Each questionnaire was assigned a number that was linked to the clinical history so that the data entered in the database would be anonymous.

The inclusion criteria were the following: (a) pregnant; (b) sufficient understanding of the Spanish language (reading, writing) to give consent and complete surveys, and (c) give birth in the hospital where the data were collected. The exclusion criteria were the following: (a) not having a complete and up-to-date history of pregnancy and childbirth and (b) having an abortion during pregnancy.

### 2.5. Instruments

#### 2.5.1. PDPI-R

The Postpartum Depression Predictors inventory includes many of the risk factors during pregnancy [19,20]. It assesses a history of previous depression, self-esteem, social support, marital satisfaction, or stressed life events, all of which are associated with pre-birth depression.

The prenatal version used in this study included 10 questions with a yes/no dichotomous answer [21]. PDPI-R has been applied to Spanish-speaking women [19] and was validated in Spain, with good internal consistency (α = 0.85), sensitivity (62.3%) and specificity (69.5%) [20].

#### 2.5.2. PHQ-9

The Patient Health Questionnaire-9 (PHQ-9)- [22,23] measures the severity of depression [24]. The DSM-5 Task Force on Depressive Disorders considers PHQ-9 as the preferred measure for depression [25]. This is a nine-item instrument with a four-level Likert scale with responses from 0 (never) to 3 (almost every day). A higher PHQ-9 score indicates a higher severity of depression symptoms (range 0–27). The recommended limit for the risk depression would be ≥10 [23,26]. It has been validated and used in different contexts and languages, including Spanish in pregnant women [27] with good internal consistency (α = 0.81). The Spanish version of the PHQ-9 depression scale was used for the study.

#### 2.5.3. Obstetric Variables

Of the obstetric variables obtained from the medical history, the following were used for this study: pregnancy complications, cesarean section, week of gestation (prematurity).

### 2.6. Data Analysis

Two groups were considered in the analyses: Spaniards and immigrants. Depression was measured with the PHQ-9 > 10. The frequency distribution was used to analyze the prevalence of qualitative variables in each group (severity levels of depressive symptoms, sociodemographic variables, social support, complications during pregnant, cesarean section and week of pregnancy at birth). Quantitative variables were expressed as mean and standard deviation. In order to analyze the differences between the groups of the different categorical variables, contingency tables were made and Pearson’s Chi-square statistics were applied, calculating Fisher’s exact test for those tables with boxes with an expected frequency of less than 5. The size of the effect was measured using Cramer’s V statistic. The confidence level was set at 95%, and the significance levels at 1% and 5% (*p* < 0.01 and *p* < 0.05). 

## 3. Results

### 3.1. Demographic Profile of the Participants

A total of 469 women participated in the study: 67% Spaniards and 33% immigrants. Of these, 26.87% were from Latin America, 4.9% from the rest of Europe, 0.4% from North America, 0.4 % from Africa, 0.2% from Asia and 0.2% from Oceania.

Table 1 describes the sociodemographic characteristics (Spaniards and immigrants). Statistically significant differences are found, although the effect size shows that the difference is weak between the two groups in level of employment (*χ*^2^_(3)_ = 11.372, *p* = 0.010, Cramer’s V= 0.156), significant difference at the educational level (*χ*^2^_(3)_ = 70.293, *p* = 0.000, Cramer’s V= 0.388), weak difference in marital status (*χ*^2^_(4)_ = 17.722, *p* = 0.001, Cramer’s V= 0.195) and primiparous mother (*χ*^2^_(1)_ = 7.200, *p* = 0.007, Cramer’s V= 0.125).

### 3.2. Depression Rates in Women Giving Birth: Severity of Depressive Symptomatology (Total of Spaniards and Immigrants)

When comparing women according the PHQ-9 questionnaire, which indicates the severity of depression symptoms, immigrant women were significantly more likely to report moderately severe depression symptoms (5.8% vs. Spaniards 1.6% (χ^2^_(1)_ = 6.364, *p* = 0.012, Cramer’s V = 0.116)) and severe symptoms (3.2% vs. Spaniards 0% (χ^2^_(1)_ = 10.238, *p* = 0.001, Cramer’s V = 0.148)) (Table 2). Although the differences are significant between both groups, the effect size shows that the association is very weak. 

### 3.3. Prevalence of Sociodemographic Factors and Symptoms of Depression by Origin

With regards to sociodemographic factors and symptoms of depression, the data show significant differences between the two groups, in all the variables except for primiparous mothers, partner problems and feeling good about living with a partner. The effect size indicates that the differences are weak, except in the lack of social support, indicating that immigrants have less social support (Table 3).

### 3.4. Types of Support at the Partner, Family and Friendship Levels

The different types of support at the partner, family and friendship levels are analyzed to see the relationship between the type of support and the origin of the women.

The data show that there were significant statistical differences in the perception of the type of support that Spaniards or immigrants have, showing a link between the lack of social support and immigrant status, although the size of the effect shows that this association is weak. There was no such relationship with partner problems (Table 4).

### 3.5. Pregnancy Complications

It was found that 16.4% of Spaniards had complications vs. 8.1% of immigrants. Of the complications experienced by the pregnant women, the significant ones in the case of immigrants were mild preeclampsia (*χ*^2^_(1)_ = 4.006, *p* = 0.045, Cramer’s V= 0.106), placenta previa (*χ*^2^_(1)_ = 5.975, *p* = 0.015, Cramer’s V= 0.129) and severe preeclampsia and IUGR (*χ*^2^_(1)_ = 4.006, *p* = 0.045, Cramer’s V= 0.106). The most common complication in Spanish women was thrombocytopenia (*χ*^2^_(1)_ = 4.150, *p* = 0.042, Cramer’s V= 0.106). Although the results were significant, the effects size was very weak. The relationship between pregnancy complications and the women’s origin is shown in Table 5.

### 3.6. Caesarean Section

Cesarean sections were performed in 16.2% of the Spanish women vs. 7.9% of the immigrant women. Of the emergency cesarean sections, 76.8% were performed in Spaniards vs. 73.7% in immigrants; there were no significant differences between the two groups. Of the total number of cesarean sections, the problems experienced by both groups are described in Table 6.

### 3.7. Week of Gestation to Delivery

With respect to the week of pregnancy in which the baby was born, significant differences were found in both groups, although the size of the effect was very weak. Immigrants have more extremely premature babies (*χ*^2^_(1)_ = 5.526, *p* =.019, Cramer’s V= 0.116), and Spaniards have more post-term babies (*χ*^2^_(1)_ = 9.505, *p* = 0.002, Cramer’s V= 0.153), shown in Table 7.

## 4. Discussion

It can be inferred from the data that Spanish women are later to have their first baby, since their average age is 34.7 and they have children up to the age 46, while immigrants, with an average of 31.7 years, have children up to 43. Immigrants start to have children at 16 and Spanish women at 18. Spanish women have longer gestation times (week 39.3 vs. 38.6 for immigrants). Spanish women are more often first-timers than immigrants. The natural population movement [28] warns of the decrease in birth rates, especially in Spanish mothers, largely due to the decrease in children per woman but also due to the reduction in the number of women of childbearing age.

There are sociodemographic variables that are considered personal antecedents for suffering emotional problems [29]. One important variable is migration. The migratory phenomenon, in which women migrate in search of an economic and social improvement or independence from their husbands [30]; because they do not have studies or have only basic education and do not have work; because of lack of support from their partner family and friends; and dissatisfaction with their relationship, cause pregnant women to sometimes lack sufficient physical and emotional support during pregnancy and post-birth [31]. 

There is international literature on risk factors [32,33] which agree that life-threatening events during pregnancy, low levels of social support and a previous history of depression are the best predictors of depression. On the contrary, strong social support for pregnant women is linked to positive mental health and better delivery outcomes. The social support perceived by pregnant women can define their level of well-being, showing an association between depression and the perception of low social, emotional and instrumental support [34], but an extensive network in precarious conditions could also increase symptomatology [34]. In the study, more Spaniards than immigrants reported having suffered symptoms of previous depression. However, there were differences between Spaniards and immigrants in the PHQ-9 score > 10 (severity of symptoms). The immigrants had more symptoms of moderately severe depression and severe symptoms than the Spaniards. In sociodemographic factors and symptoms of depression, more immigrants than Spaniards did not have a partner, had basic or secondary education, did not have a job, perceived they did not have social support (partner, family and friends) and had more dissatisfaction in their relationship with their partner. For the immigrants, the lack of emotional support, level of trust, being unable to count on others and lack of practical support in terms of partner, family and friendship support were significant risk factors.

With respect to obstetric complications, there were no differences in most variables. Spanish women had complications of thrombocytopenia. Immigrant women had problems with placenta previa; preeclampsia, both mild and severe; and IUGR. Important female health problems such as hypertension should be monitored. It is important to clearly identify and assess the prevalence of depressive and anxiety disorders in hypertensive women and to determine the contribution that each of these emotional disorders can have to the prediction of perceived symptoms, which can be considered as an indicator of physical and psychological ill-health [35,36].

In relation to sociocultural and obstetric factors, there are contradictory studies both nationally and internationally in which the lower cultural and educational level of pregnant women is associated with poorer identification of the relevant symptomatology, not perceiving the risk. In our sample, there were significant differences between Spanish and immigrant women in terms of educational level, absence of work and marital status, which could be related to adverse obstetric outcomes, but this is not the case. There were more Spaniards than immigrants who had pregnancy complications and underwent cesarean section. These data are consistent with other studies where the risk of poor prenatal evolution is greater in native women than in immigrant women. This may be because immigrants develop healthier behaviors, such as consuming less tobacco and alcohol and having healthy diets [37]. Another possible explanation for this result is aligned with the “healthy migrant effect,” the belief that immigrants are healthier than natives, overcoming the stress of leaving home to settle in a new country [38].

As for cesarean section, in recent decades, cesarean deliveries have been systematized in many countries and should not exceed 15% due to the risks that this entails [37]. The results obtained show that more cesarean sections are performed on Spaniards than on immigrants, with a higher percentage of Spanish women having an emergency cesarean section than immigrants, but the data are not significant. These data are in line with another investigation [9] in which no significant differences were found between both groups, coinciding with other studies both nationally and internationally [9]. This absence of differences is due to the universality of the Spanish Health System, where the Bioethical Principle of Justice or Equity exists, as well as the existence of very solid cesarean section protocols for all [39] so that attempts are made to perform the strictly necessary ones.

As regards prematurity, a report on births is published annually in the United States [40]. This report shows the differences each year in prematurity by origin. These differences persist regardless of the socioeconomic level of pregnant women or their degree of access to health insurance. The results of this investigation show that more immigrants have extremely premature and very premature babies, while more Spanish women have a post-term baby. There are no data on this relationship, so it is difficult to know whether it is a physical question or related to psychological criteria of fear of consulting and making emergency visit vs. having security in the system, nor is it known whether it may be due to nutritional mechanisms. Further research is needed.

As limitations to the study, we find that all foreign women are not a homogeneous group, so classifying them into the same group can lead to biases. (a) All of them were selected for this research because they had linguistic facility with Spanish. (b) This study was carried out only in one hospital in Spain. In future studies, the same analyses should be extended to other hospitals to ensure the generalizability of the sample. (c) Inferences drawn from this paper should be treated with caution until more research is conducted to corroborate the findings.

Despite these limitations, the strength of this study is the large amount of data for the specified variables within this analysis.

## 5. Conclusions

Depression in pregnancy is a risk factor for many problems, such as late or delayed antenatal care and self-neglect [33], which can cause problems in the child’s development during pregnancy [32,33] and after birth [4,33].

There continues to be care based on the biomedical model to treat depressive symptoms, as they are considered a mental illness. It is important that the therapies used by professionals connect with what the person needs, that the model used takes into account the psychosocial nature of this problem and the importance of the role played by emotions or learning in the development of depressive symptoms. It is important to keep in mind the dynamic, contextual, holistic, interactive and functional nature of psychological disorders [41]. The literature shows that women prefer psychological interventions in the treatment of perinatal depression, to pharmacological interventions [4]. Screening and interventions for depression during pregnancy must take into account immigration status for healthcare to be effective. Immigration status can have different risk factors that affect depression during pregnancy [7].

Although there is a European commitment to address inequalities by providing quality health care, which meets the needs of immigrant women of childbearing age, it is not provided and financed in the same way in all countries; some provide free care, others require health insurance, and others are available to those who make contributions to national insurance through work [8].

This study could indicate that the relationship between nationality and psychological and psychosocial conditions with pregnancy has an impact on their obstetric results, with gestational health being worse in immigrant women than in Spanish women because they have these better sociodemographic conditions, but this is not always the case. In Spain, access to universal healthcare is guaranteed for pregnant women, whether native, legal or illegal immigrants, and hence access to Spanish hospitals where universal care exists causes them to have the same care and therefore greater control both in pregnancy and at the time of birth. This situation favors compliance with the Bioethical Principles of Autonomy, Beneficence, Non-maleficence and Justice.

One of the strengths of this study is that it provides data on one of the problems that has been worrying health professionals in different fields such as obstetrics, nursing, pediatrics and psychology for a few years. In our area of knowledge, there are few studies on the subject, and to the best of our knowledge, there are no national studies that analyze objective tests (obstetric history of pregnancy and childbirth) and the study of psychosocial and psychological variables to see how they affect obstetric health, comparing Spaniards with immigrants. It is about identifying population and risk factors to improve clinical applicability, maintaining and promoting its positive potential for maternal and perinatal health.

To this end, in relation to immigration, the Spanish health system should promote mediators who advise and accompany immigrant women who do not speak Spanish, so that they can carry out the procedures and medical consultations related to their pregnancy, translating, interpreting and informing them of the existing resources, allowing the mothers to make their own decisions. For this reason, support should be given to health professionals when assisting immigrant women. 

## Figures and Tables

**Figure 1 ijerph-19-11782-f001:**
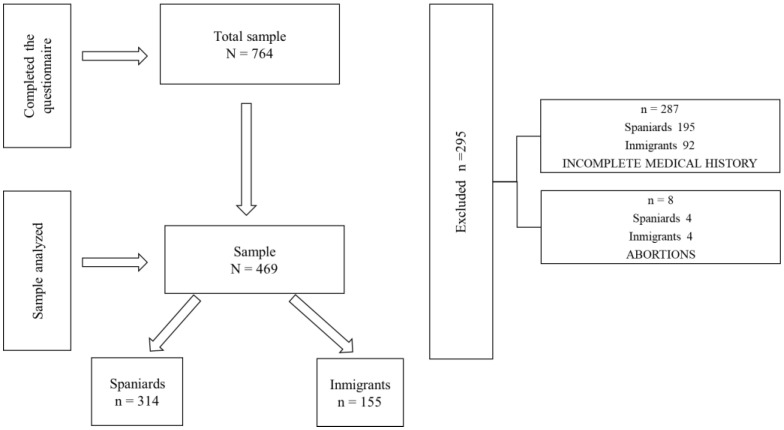
Sample of participants.

**Table 1 ijerph-19-11782-t001:** Sociodemographic characteristics.

	Spaniards *n* (314) 67%		Inmigrants *n* (155) 33%			
	M	SD	M	SD	*t*	
**Age**	34.72	4.77	31.74	5.63	5.997 **	
**Week of pregnancy**	39.32	1.8	38.65	2.72	3.159 **	
	n	%	n	%	*χ* ^2^	Cramer’s V
**Employment**					11.372 *	0.156
Active	250	80.1	103	67.3		
Unemployed	15	4.8	18	11.8		
Housewives	46	14.8	31	20.3		
Disability	1	0.3	1	0.7		
**Level of study**					70.293 **	0.388
No studies	4	1.3	6	3.9		
Basics	28	9	20	12.9		
Medium-high	74	23.7	89	57.4		
University	206	66	40	25.8		
**Marital status**					17.722 **	0.195
Married	163	52.1	67	43.8		
Single	34	10.9	38	24.8		
Live with a partner	114	36.4	45	29.4		
Widow	1	0.3	2	1.3		
Separated/divorced	1	0.3	1	0.7		
**Smoking**					3.067	
Yes	34	18	7	9.3		
No	155	82	68	90.7		
**Drinking alcohol**					1.046	
Yes	5	3	4	5.8		
No	162	97	65	94.2		
**Primiparous mothers**					7.200 **	0.125
Yes	151	48.9	55	35.7		
No	158	51.1	99	64.3		
**Previous depression**					1.832	
Yes	11	3.6	2	1.3		
No	297	96.4	148	98.7		

* *p* < 0.05, ** *p* < 0.01.

**Table 2 ijerph-19-11782-t002:** Differences in severity of symptoms if they are Spaniards or immigrants.

	Spaniards *n*		Immigrants *n*			
	M	SD	M	SD	*t*	
**Depression Symptoms**	0.18	0.425	0.35	0.719	−3.042 **	
**PHQ-9 Severity**	** *n* **	**%**	** *n* **	**%**	** *χ* ** ** ^2^ **	**Cramer’s V**
No symptoms of depression (<10)	264	84.1%	119	76.8%	3.695	
Moderate depression symptoms (10–14)	45	14.3%	23	14.8%	0.022	
Moderately severe symptoms (15–19)	5	1.6%	9	5.8%	6.364 *	0.116 *
Severe symptoms (≥20) Major depression epression	0		5	3.2%	10.238 **	0.148 *

* *p* < 0.05, ** *p* < 0.01. Note: PHQ-9 Patient Health Questionnaire.

**Table 3 ijerph-19-11782-t003:** Prevalence of sociodemographic factors and symptoms of depression by origin.

PHQ-9		Spaniards		Immigrants		*χ* ^2^	Cramer’s V
		*n*	%	*n*	%		
**Partner**	With partner	41	13.8	19	13.7	17.450 **	0.200
	No partner	7	2.3	13	9.3		
**Studies**	No studies/ Basics	39	13.1	25	17.9	16.306 **	0.193
	Medium/ Higher	9	3.1	7	5		
**Employment** **situation**	Employed	33	11	15	9.7	18.886 **	0.208
	Unemployed	11	5	17	12.2		
**Previous depression**		4	1.3	1	0.7	28.054 **	0.256
**Primiparous mothers**		25	8.5	7	5	1.262	
**Lack of partner support**		3	1	8	5.9	13.909 **	0.181
**Lack of family support**		5	1.7	10	7.3	49.150 **	0.338
**Lack of friendly support**		5	1.7	14	10.4	17.631 **	0.204
**Partner dissatisfaction**		2	0.8	7	5.5	8.936 *	0.147
Problems with the partner		4	1.5	3	2.4	4.861	
They don’t feel good in the relationship		45	15.7	23	17.9	6.871	

* *p* < 0.05, ** *p* < 0.01.

**Table 4 ijerph-19-11782-t004:** Lack of social support according to origin.

Social Support	Spaniards		Immigrants		*χ* ^2^	Cramer’s V
	*n*	%	*n*	%		
**Lack of partner support**						
Emotional	20	6.6	20	13.4	5.709 *	0.113
Trust	12	4	14	9.5	5.583 *	0.112
Count on the partner	12	4	14	9.5	5.447 *	0.110
Practical support	23	7.6	22	15.1	6.050 *	0.116
**Lack of family support**						
Emotional	15	4.9	16	10.8	5.428 *	0.109
Trust	11	3.6	12	8.1	4.081 *	0.095
Count on the family	10	3.3	15	10.1	8.809 **	0.139
Practical support	32	10.6	32	21.8	10.191 **	0.150
**Lack of friendly support**						
Emotional	11	3.6	14	9.5	6.505 *	0.120
Trust	9	3	27	18.6	32.079 **	0.268
Count on friends	15	5	30	20.5	26.170 **	0.242
Practical support	54	18.1	49	34.5	14.548 **	0.182
**Relationship with partner**						
Dissatisfaction with the relationship	12	4.1	16	11.3	8.413 **	0.139
Partner problems	20	6.7	10	7.1	0.019	
Not well in partner	17	5.7	16	11.3	4.384 *	0.100

* *p* < 0.05, ** *p* < 0.01.

**Table 5 ijerph-19-11782-t005:** Descriptive complications during pregnancy.

Descriptive Complications	Spaniards		Immigrants		*χ* ^2^	Cramer’s V
	*n*	%	*n*	%		
High obstetric risk	13	3.5	2	0.5	2.561	
Intrauterine Growth, Restriction (IUGR) TYPE I	6	1.7	1	0.3	1.099	
IUGR TYPE II	1	0.3	0		0.493	
HTA high	1	0.3	2	0.6	1.513	
Mild preeclampsia	0		2	0.6	4.006 *	0.106
Severe preeclampsia	2	0.6	2	0.6	0.512	
Suspicion of threat of early birth labor	3	0.8	1	0.3	0.116	
Placenta praevia	0		3	0.8	5.975 *	0.129
Cervical shortening with pessary	4	1.1	2	0.6	0.000	
Cervical shortening without pessary	1	0.3	0		0.493	
High fetal cardiological risk	1	0.3	0		0.493	
Vaginal or urinary infections	2	0.6	1	0.3	0.000	
Poor obstetric control	1	0.3	3	0.8	3.123	
Fetus small for gestational age	1	0.3	1	0.3	0.258	
Suspected macrosomic fetus	1	0.3	1	0.3	0.258	
Thrombocytopenia	13	5.2	1	0.3	4.150 *	0.106
High risk of Down’s Syndrome	2	0.6	0	0	0.985	
Syphilis	0		1	0.8	2.014	
Repeated miscarriages and hypertransaminases	1	0.3	0		0.493	
Severe preeclampsia and Intrauterine Growth Restriction (IUGR)	0		2	0.6	4.006 *	0.106

* *p* < 0.05.

**Table 6 ijerph-19-11782-t006:** Caesarean section.

Reason for C-Section	Spaniards		Immigrants		*χ* ^2^
	*n*	%	*n*	%	
Cephalic pelvic disproportion	12	3.2	2	0.5	2.187
Podalic presentation	3	0.8	0		1.481
Polyhydramnios, transverse lie, uterine hypotonia and high risk of Down’s Syndrome	1	0.3	0		0.495
No progression of labor	13	3.5	6	1.6	0.020
Suspected fetal compromise	22	7.7	23	5.8	3.259
Failed labor induction	7	1.9	3	0.8	0.044
Requested by the patient	2	0.6	1	0.3	0.000
Placenta praevia	4	1.1	2	0.6	0.000
Placental abruption	5	1.4	1	0.3	0.726
Iterative	1	0.3	2	0.6	1.502
Iterative with tubal ligation	1	0.3	3	0.8	3.105
Breech presentation	2	0.6	0		0.989
Non-reassuring fetal heart rate, sustained fetal bradycardia, loss of fetal well-being	1	0.3	1	0.3	0.255
Difficult extraction of the fetus, very ascended vesicouterine plica, requires dissection 1 with a swab.	1	0.3	0		0.495
Urgent caesarean section with LMWH prophylaxis (anticoagulant in pregnancy)	1	0.3	0		0.495
IUGR type II	1	0.3	0		0.495
Fetal maternal interest	1	0.3	1	0.3	0.255
Scheduled caesarean section	2	0.3	0		0.989
Twin gestation (diamniotic dichorionic, scheduled caesarean section for breech-cephalic presentation)	1	0.3	0		0.495
Preterm, premature rupture of membranes, tocolysis with Atosiban	1	0.3	0		0.495
Placenta praevia, transverse lie	0		1	0.3	2.006
Fetal pathology	1	0.3	0		0.495
Due to risk of the first baby and feticide of the second fetus, pelvic-cephalic disproportion, premature rupture of the membranes at term	1	0.3	0		0.495
Pelvic-cephalic disproportion, premature rupture of membranes at term	1	0.3	0		0.495
Severe preeclampsia with macrosomia	0		1	0.3	2.006
Metrorrhagia, maternal anemia, suspected placental abruption and fetal malposition	1	0.3	0		0.495

**Table 7 ijerph-19-11782-t007:** Week of pregnancy at birth.

Week of Pregnancy at Birth	Spaniards		Immigrants		*χ*2	Cramer’s V
	*n*	%	*n*	%		
Extremely premature (20–28 weeks)	0		3	2	5.526 *	0.124
Very premature (28–32 weeks)	0		2	1.3	3.703	
Moderately premature (32–37 weeks)	29	9.5	16	10.5	0.009	
At term (37–41 weeks)	232	75.8	124	81.6	2.122	
Post-term (41–42 weeks)	45	14.7	7	4.6	9.505 **	0.153

* *p* < 0.05, ** *p* < 0.01.

## Data Availability

Not applicable.

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
