# Peer review of "Psychological, Psychosocial and Obstetric Differences between Spanish and Immigrant Mothers: Retrospective Observational Study"

_ijerph, 2022, doi:10.3390/ijerph191811782_

Round 1

Reviewer 1 Report

You mentioned this as an retrospective observational study which means that all cases have already happened before the study begins. Please clarify phrase below or change title.

" The questionnaire was completed individually when the women 114 attended obstetric consultations at 12 weeks of pregnancy for the first-trimester ultra- 115 sound examination." 

When you use PHQ-9 - if obstetric variables were collected from questionnaire than some informations could be biased (such as dystocic delivery) due to inconsistent knowledge of medical terms by patients.

line 183 - "and feeling good in a partner". Please clarify

line 200 and table 5 - "previous placenta" - Do you refer to placenta praevia or previous placental discorders?

Table 6 - loss of fetal well being is a general term and it cannot be quantified. In obstetrics we normaly use suspected fetal compromise.

Table 7 - CS  Requested by the husband, woman with anxiety disorder. 

This is unacceptable medical situation as well as a mention in a scientifical article.

Results section should be extensively revised excluding incorrect obstetrical cs indications and associated informations. 

Discussion section should focus on the discovered issue - foreigners are more predisposed to adverse outcome due to psychological factors and a conclusion on how can the Spanish system improve the outcome of the foreign women.

Reviewer 2 Report

Congratulations to the authors for this important study. The methodology was sound and well conducted.

My concerns relate to the inferences draw from the findings of higher levels of psychological distress and obstetric outcomes. The authors infer that problems with partners may contribute to stress but do not point out that the migrant women had higher levels of social support. I was surprised that possible confounders such as health literacy, poorer dietary intake, and models of care were not discussed. The authors state the Spain has a universal care system. Whilst his provides access, there was no discussion of the cultural competency of the system. It is possible that the lack of cultural responsiveness of the system is a contributer to lower levels of antenatal care. I am also concerned about the implications of the paper on care. Are the authors recommending greater screening for depression in migrant women? Why is there no suggestion of follow up qualitative research surfacing the patient experience of care. The limitations such should note that inferences draw from this paper should be treated with caution until further research is conducted to explore the lived experience. I recommend publication with the issues I have raised addressed.

Round 2

Reviewer 1 Report

Table 6 has multiple non-suitable items (please revise throughly or exclude) -

No Problem - if no problem than why is included at dystocic delivery?

Bag break - I presume that you refer to ROM (Rupture of membranes), this is not a reason for dystocic delivery only by itself

Low reassuring - change with non-reassuring fetal heart rate (please see FIGO CTG guidelines for intrapartum monitoring)

Induction for non-progessing of labor is correct instead of childbirth

Instrumentation for expulsion relief - rephrase with intrumental delivery for maternal exhaustion

Caesarean due to lack.. - lack of what? please revise or exclude

Pathology of the umbilical... cord, I presume. Please add

Baby death is a commom language not professional language - change with fetal demise

Table 7 - if a patient had hysterectomy than is impossible to perform a c section without a uterus. Please clarify - change with hysteroscopy or exclude.
“Difficult extraction of the fetus, previous hysterectomy, requires dissection 1 with a swab"

Potalic presentation - Is Podalic presentation or breech which was correct.

Transverse situation - is transverse lie

No progression of labor was correct

Broken bag - same with table 6

Fetus problems - change with fetal pathology or exclude

Severe preeclapmsia big boy - change with severe preeclampsia with macrosomia
